# Fairness Continual Learning Approach to Semantic Scene Understanding in Open-World Environments

**Thanh-Dat Truong**[1]**, Hoang-Quan Nguyen**[1]**, Bhiksha Raj**[2,3]**, and Khoa Luu**[1]

[1]CVIU Lab, University of Arkansas, Fayetteville, AR, 72701
[2]Carnegie Mellon University, Pittsburgh, PA, 15213
[3]Mohammed bin Zayed University of AI, Abu Dhabi, UAE
{tt032, hn016, khoaluu}@uark.edu, bhiksha@cs.cmu.edu

## Abstract

Continual semantic segmentation aims to learn new classes while maintaining the information from the previous classes. Although prior studies have shown impressive progress in recent years, the fairness concern in the continual semantic segmentation needs to be better addressed. Meanwhile, fairness is one of the most vital factors in deploying the deep learning model, especially in human-related or safety applications. In this paper, we present a novel Fairness Continual Learning approach to the semantic segmentation problem. In particular, under the fairness objective, a new fairness continual learning framework is proposed based on class distributions. Then, a novel Prototypical Contrastive Clustering loss is proposed to address the significant challenges in continual learning, i.e., catastrophic forgetting and background shift. Our proposed loss has also been proven as a novel, generalized learning paradigm of knowledge distillation commonly used in continual learning. Moreover, the proposed Conditional Structural Consistency loss further regularized the structural constraint of the predicted segmentation. Our proposed approach has achieved State-of-the-Art performance on three standard scene understanding benchmarks, i.e., ADE20K, Cityscapes, and Pascal VOC, and promoted the fairness of the segmentation model.

## 1 Introductions

Convolutional Neural Networks (CNNs) [5, 7] and Transformers [45, 8] have been introduced to approach semantic segmentation tasks where the models learn from the large-scale data having known classes at once. These segmentation models learned on large-scale data may perform poorly as they may counter the new objects or new environments. To bridge this gap, several approaches have been proposed to adapt the model to the new data. Domain Adaptation is one of the common approaches [39, 42, 43, 1, 18] that adaptively transfer the segmentation model into the deployed environment. However, domain adaptation cannot handle when new objects appear due to its close-set setup. Also, this approach requires access to both original and new training data. In practice, the segmentation models should be capable of learning new classes continually without re-training from the previous data. This paradigm is defined as Continual Semantic Segmentation

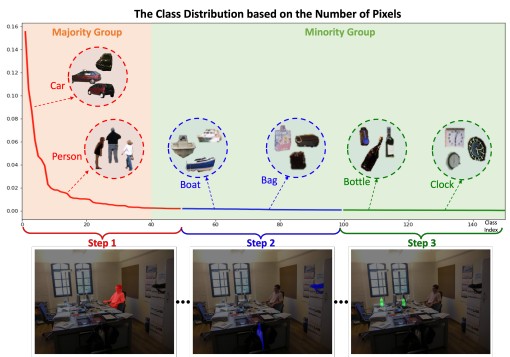

Figure 1: **The Class Distribution of ADE20K.** In the ADE20K 50-50 (3 steps) benchmark, the distribution of classes in the majority group in the early step dominates the ones in the minority groups in the later steps. The distributions of classes gradually decrease over steps.

37th Conference on Neural Information Processing Systems (NeurIPS 2023).

[12, 46]. The current continual semantic segmentation approaches [12, 46] concentrate on addressing two main challenges, i.e., (1) catastrophic forgetting [33, 14, 38] and (2) background shift [12, 46]. The former problem indicates the forgetting issue of the model about the previous knowledge when learning on new training data [12, 46, 4]. Meanwhile, the latter problem refers to the classes of previous or future data that have collapsed into a background class [3]. However, another critical problem that needs attention is the fairness issue in semantic segmentation.

Fairness in semantic segmentation refers to the problem of the model where it behaves unfairly between classes in the dataset due to the class distribution, i.e., the model tends to produce prediction bias toward a specific group of classes occupying a large region on an image or frequently existing in the dataset (as shown in Fig. 1). The unfair predictions could result in severe problems, especially in human-related applications that could influence human safety. Moreover, the fairness problem could even be well observed in the context of continual learning when the model encounters new classes without accessing previous training data. The prior work of continual learning in image classification [48] has also considered this problem. Several prior studies [1, 18, 19] in semantic segmentation have tried to reduce the effect of the class imbalance by introducing the weighted cross entropy [11, 44, 19], focal loss [1], over-sampling techniques [18, 47, 1]. However, the fairness problem in continual semantic segmentation has yet to be well-defined and directly addressed. There should be more attention on addressing the fairness issue in continual semantic segmentation. Therefore, this work aims to address the fairness problem in continual semantic segmentation caused by the imbalanced class distribution defined based on the number of pixels of each class in the dataset.

**Contributions of this Work:** This work presents a novel **Fair**ness **C**ontinual **L**earning (**FairCL**) approach to semantic scene segmentation. Our contributions can be summarized as follows. First, under the perspective of fairness learning, the new metric is formulated to measure the fairness of the model via the error rate among classes. Then, the metric is further derived into the three main objectives, i.e., (1) the *Task-specific Objective* that handles the catastrophic forgetting problem, (2) the *Fairness Objective* that maintains the fairness of predictions produced by the model based on the class distribution, and (3) the *Conditional Structural Constraint* that imposes the consistency of the segmentation predictions. Second, to sufficiently model the continual learning problem, the novel *Prototypical Contrastive Clustering* loss is presented to address the catastrophic forgetting and the background shifting problems. Moreover, the proposed Prototypical Contrastive Clustering loss has been proven to be a generalized paradigm of knowledge distillation approaches commonly adopted in continual learning. Last, sufficient ablation studies have shown the effectiveness of our method in continual learning and promoted the fairness of the model. The empirical comparison with prior methods has shown the State-of-the-Art (SOTA) results on the standard benchmarks.

## 2 Related Work

**Continual Semantic Segmentation** While there is enormous progress in the continual learning problem, most approaches focus on image classification. Existing continual learning methods [22, 17] have been extended for continual semantic segmentation in medical images [27, 28] and general datasets [24]. Cermelli et al. [3] addressed a background shifting problem in the continual semantic segmentation. Douillard et al. [12] introduced a multi-scale spatial distillation scheme that preserves long- and short-range spatial relationships at the feature level. Volpi et al. [41] presented a continual learning framework where the model is sequentially trained on multiple labeled data domains. Rostami et al. [34] proposed a continual learning framework under the unsupervised domain adaptation setting using data rehearsal. Saporta et al. [35] introduced a multi-head knowledge distillation framework for the continual unsupervised domain. Zhang et al. [46] presented a representation compensation module to decouple the representation learning of both old and new knowledge. Cha et al. [4] suggested finding the unknown class from the background to distinguish the representations of the potential future classes. Qiu et al. [30] proposed a self-attention transferring method to capture both within-class and between-class knowledge. Phan et al. [29] introduced a class-similarity knowledge-distillation method to revise the old classes more likely to be forgotten and better learn new classes related to the previous classes.

**Class Imbalance and Fairness Approaches in Semantic Segmentation** Jiawei et al. [32] presented a balanced Softmax loss that reduces the distribution shift of labels and alleviates the long-tail issue. Wang et al. [44] proposed a Seesaw loss that reweights the contributions of gradients produced by positive and negative instances of a class by using two regularizers, i.e., mitigation and compensation. Liu et al. [23] proposed an algorithm that handles imbalanced classification, few-shot learning, and

open-set recognition using dynamic meta-embedding. Chu et al. [9] proposed a stochastic training scheme for semantic segmentation, which improves the learning of debiased and disentangled representations. Szabo et al. [37] proposed tilted cross-entropy loss to reduce the performance differences, which promotes fairness among the target classes. Truong et al. [40] introduced a fairness domain adaptation approach to semantic segmentation that maintains the fairness of the predictions produced by the segmentation model on the target domain.

## 3 The Proposed Fairness Continual Learning Approach

Let $\mathcal{F}$ parameterized by $\theta$ be the deep semantic segmentation model that maps an input image $\mathbf{x} \in \mathcal{X}$ to the segmentation map $\mathbf{y} \in \mathcal{Y}$, $\mathbf{y} = \mathcal{F}(\mathbf{x}, \theta)$. Continual Semantic Segmentation (CSS) aims to learn a model in $T$ steps. In each step, the segmentation model $\mathcal{F}$ encounters a dataset $\mathcal{D}^t = \{\mathbf{x}^t, \hat{\mathbf{y}}^t\}$ where $\mathbf{x}^t$ is the input image and $\hat{\mathbf{y}}^t$ is the ground-truth segmentation at time $t \in [1..T]$. The current ground-truth segmentation map $\hat{\mathbf{y}}^t$ only contains the labels of the current classes $\mathcal{C}^t$ and all the class labels of prevision steps, $\mathcal{C}^{1..t-1}$, or the future steps, $\mathcal{C}^{t+1..T}$ are collapsed into a background class or ignored. Formally, learning the semantic segmentation at time step $t$ can be formulated as follows:

$$\theta_t^* = \arg\min_{\theta_t} \mathbb{E}_{\mathbf{x}^t, \hat{\mathbf{y}}^t \in \mathcal{D}^t} \mathcal{L}\left(\mathcal{F}(\mathbf{x}^t, \theta_t), \hat{\mathbf{y}}^t\right) \tag{1}$$

where $\theta_t^*$ is the parameters of $\mathcal{F}$ at time step $t$, $\mathcal{L}$ is the objective learning of the continual learning task. In CSS learning, at the current time step $t$, the segmentation model $\mathcal{F}$ is expected to not only predict all the classes $\mathcal{C}^{1..t-1}$ learned in the previous steps but also predict the current new classes $\mathcal{C}^t$. Three significant challenges have been identified in this learning setting and should be addressed.

- **Background Shift** At time step $t$, the labels of previous and future steps have been ignored. Thus, the pixels of these classes are ambiguous, which means these could contain either the class of previous or future steps. During learning $\mathcal{C}^t$, the model could consider these classes as negative samples. As a result, the model tends to learn non-discriminative features for these pixels, leading to difficulty learning new classes or forgetting the old ones.

- **Catastrophic Forgetting** cause the model may partially or completely forget the knowledge of classes $\mathcal{C}^{1..t-1}$ when learning the new classes $\mathcal{C}^t$. This problem could be caused by the background shift and the learning mechanism. Since classes in $\mathcal{C}^{1..t-1}$ are considered as the background class at time step $t$, the model tends to update the knowledge of the new classes while the predictions of classes incline to be suppressed.

- **Fairness** While the prior approaches [12, 46, 4] focus on addressing the two above challenges, the fairness issue has received less attention and has not been well addressed yet. However, fairness is one of the most important criteria as it guarantees the model behaves fairly among not only classes in $\mathcal{C}^t$ but also classes in $\mathcal{C}^{1..t}$ that have been learned. The fairness in CSS is typically caused by the imbalance distribution between classes as several classes occupy the larger portion or exist more frequently than other classes (Fig. 1). In addition, the appearance of training samples at the training step $t$ also exaggerates the bias due to the new classes since the classes in the previous training steps have collapsed.

To address these challenges in CSS, we first reconsider the semantic segmentation problem from the fairness viewpoint followed by introducing a novel approach to alleviate the fairness issue based on the ideal class distribution. Then, we introduce a novel Prototypical Contrastive Clustering loss to model background classes and catastrophic forgetting.

### 3.1 Fairness Learning Approach to Continual Semantic Segmentation

Maintaining fairness is one of the most important factors in learning CSS as it reduces the bias of the model toward a certain group of classes. Formally, under the fairness constraint in semantic segmentation, the error rate of each class should be equally treated by the segmentation model. Given classes $c_a$ and $c_b$ in the set of classes $\mathcal{C}^{1..T}$, this constraint can be formulated as follows:

$$\max_{c_a, c_b \in \mathcal{C}^{1..T}} \left| \mathbb{E}_{\mathbf{x}, \hat{\mathbf{y}} \in \mathcal{D}} \sum_{i,j} \mathcal{L}\left(y_{i,j}, \hat{y}_{i,j} = c_a\right) - \mathbb{E}_{\mathbf{x}, \hat{\mathbf{y}}} \sum_{i,j} \mathcal{L}\left(y_{i,j}, \hat{y}_{i,j} = c_b\right) \right| \leq \epsilon \tag{2}$$

where $y_{i,j}$ and $\hat{y}_{i,j}$ is the prediction and ground truth at the pixel location $(i, j)$, respectively; the loss function $\mathcal{L}$ measures the error rates of predictions. Intuitively, Eqn. (2) aims to maintain the

differences in the error rates between classes lower than a small threshold $\epsilon$ to guarantee fairness among classes. For simplicity, this constraint can be considered as an additional loss while optimizing Eqn. (1). However, this approach could not guarantee the model being fair due to Eqn. (3).

$$
\begin{aligned}
&\max_{c_a, c_b \in \mathcal{C}^{1..T}} \left| \mathbb{E}_{\mathbf{x}, \hat{\mathbf{y}} \in \mathcal{D}} \sum_{i,j} \mathcal{L}\left(y_{i,j}, \hat{y}_{i,j} = c_a\right) - \mathbb{E}_{\mathbf{x}, \hat{\mathbf{y}}} \sum_{i,j} \mathcal{L}\left(y_{i,j}, \hat{y}_{i,j} = c_b\right) \right| \\
&\leq \sum_{c_a, c_b \in \mathcal{C}^{1..T}} \left| \mathbb{E}_{\mathbf{x}, \hat{\mathbf{y}} \in \mathcal{D}} \sum_{i,j} \mathcal{L}\left(y_{i,j}, \hat{y}_{i,j} = c_a\right) - \mathbb{E}_{\mathbf{x}, \hat{\mathbf{y}}} \sum_{i,j} \mathcal{L}\left(y_{i,j}, \hat{y}_{i,j} = c_b\right) \right| \leq 2 \left| \mathcal{C}^{1..T} \right| \left[ \mathbb{E}_{\mathbf{x}, \hat{\mathbf{y}} \in \mathcal{D}} \mathcal{L}\left(\mathcal{F}(\mathbf{x}, \theta), \hat{\mathbf{y}}\right) \right]
\end{aligned}
\tag{3}
$$

As shown in the above inequality, the constraint of Eqn. (2) has been bounded by the loss function in Eqn. (1). Although minimizing Eqn. (1) could also impose the constraint in Eqn. (2), the fairness issue still remains unsolved due to the imbalanced class distributions indicated through Eqn. (4).

$$
\theta^* = \arg \min_\theta \left[ \int \mathcal{L}(\mathbf{y}, \hat{\mathbf{y}}) p(\mathbf{y}) p(\hat{\mathbf{y}}) d\mathbf{y} d\hat{\mathbf{y}} \right] = \arg \min_\theta \left[ \int \sum_{i,j} \mathcal{L}(y_{i,j}, \hat{y}_{i,j}) p(y_{i,j}) p(\mathbf{y}_{\backslash(i,j)} | y_{i,j}) p(\hat{\mathbf{y}}) d\mathbf{y} d\hat{\mathbf{y}} \right]
\tag{4}
$$

where $p(\cdot)$ is the data distribution, $\mathbf{y}_{\backslash(i,j)}$ is the predicted segmentation map without the pixel at position $(i, j)$, $p(y^k)$ is the class distribution of pixels, and $p(\mathbf{y}_{\backslash(i,j)} | y_{(i,j)})$ is the conditional structural constraint of the predicted segmentation $\mathbf{y}_{\backslash(i,j)}$ conditioned on the prediction $y_{i,j}$. Practically, the class distributions of pixels $p(y_{i,j})$ suffer imbalanced issues, where several classes in the majority group significantly dominate other classes in the minority group. Then, learning by gradient descent method, the model could potentially bias towards the class in the majority group because the produced gradients of classes in the majority group tend to prevail over the ones in the minority group. Formally, considering the two classes $c_a$ and $c_b$ of the dataset where their distributions are skewed, i.e., $p(c_a) < p(c_b)$, the gradients produced are favored towards class $c_b$ as shown in Eqn. (5).

$$
\begin{aligned}
&\left\| \frac{\partial \int \sum_{i,j} \mathcal{L}(y_{i,j}, \hat{y}_{i,j}) p(y_{i,j} = c_a) p(\mathbf{y}_{\backslash(i,j)} | y_{i,j}) p(\hat{\mathbf{y}}) d\mathbf{y} d\hat{\mathbf{y}}}{\partial \mathbf{y}_{(c_a)}} \right\| \\
&< \left\| \frac{\partial \int \sum_{k=1}^N \mathcal{L}(y_{i,j}, \hat{y}_{i,j}) p(y_{i,j} = c_b) q(\mathbf{y}_{\backslash(i,j)} | y_{i,j}) p(\hat{\mathbf{y}}_k) d\mathbf{y} d\hat{\mathbf{y}}}{\partial \mathbf{y}_{(c_b)}} \right\|
\end{aligned}
\tag{5}
$$

where $|| \cdot ||$ is the magnitude of gradients, $\mathbf{y}_{(c_a)}$ and $\mathbf{y}_{(c_b)}$ are the predictions of classes $c_a$ and $c_b$.

**Learning Fairness from Ideally Fair Distribution** To address this problem, we first assume that there exits an ideal distribution $q(\cdot)$ where the class distributions $q(y_{i,j})$ are equally distributed. Under this assumption, the model learned is expected to behave fairly among classes as there is no bias toward any groups of classes. *It should be noted that our assumption is used to derive our learning objective and is going to be relaxed later. In other words, the ideal data is not required at training.* Then, our learning objective in Eqn. (4) could be rewritten as follows:

$$
\theta^* = \arg \min_\theta \left[ \mathbb{E}_{\mathbf{x} \sim p(\mathbf{y}), \hat{\mathbf{y}} \sim p(\hat{\mathbf{y}})} \sum_{i,j} \mathcal{L}(y_{i,j}, \hat{y}_{i,j}) \frac{q(y_{i,j}) q(\mathbf{y}_{\backslash(i,j)} | y_{i,j})}{p(y_{i,j}) p(\mathbf{y}_{\backslash(i,j)} | y_{i,j})} \right]
\tag{6}
$$

The fraction between the ideal distribution $q(\cdot)$ and the actual data distribution $p(\cdot)$ is the residual learning objective for the model to achieve the desired fairness goal. Let us further derive Eqn. (6) by taking the log as follows:

$$
\theta^* = \arg \min_\theta \mathbb{E}_{\mathbf{x} \sim p(\mathbf{x}), \hat{\mathbf{y}} \sim p(\hat{\mathbf{y}})} \left\{ \mathcal{L}(\mathbf{y}, \hat{\mathbf{y}}) + \frac{1}{N} \sum_{i,j} \left[ \log \left( \frac{q(y_{i,j})}{p(y_{i,j})} \right) + \log \left( \frac{q(\mathbf{y}_{\backslash(i,j)} | y_{i,j})}{p(\mathbf{y}_{\backslash(i,j)} | y_{i,j})} \right) \right] \right\}
\tag{7}
$$

The detailed derivation of Eqn. (6) and Eqn. (7) will be available in the supplementary. As shown in the Eqn. (7), there are three learning objectives as follows:

- **The Continual Learning Objective** The first term, i.e., $\mathcal{L}(\mathbf{y}, \hat{\mathbf{y}})$, represents the task-specific loss which is the continual learning objective. This objective aims to address the catastrophic forgetting and background shift problems. To achieve this desired goal, we introduce a novel Prototypical Contrastive Clustering Loss that will be discussed in Sec. 3.2.

- **The Fairness Objective** The second term, i.e., $\mathcal{L}_{class} = \log \left( \frac{q(y_{i,j})}{p(y_{i,j})} \right)$, maintains the fairness in the predictions produced by the model. This objective penalizes the prediction of

Figure 2: **The Proposed Fairness Continual Learning Framework.** The predicted segmentation maps are imposed the cross-entropy loss, the Prototypical Contrastive Clustering loss ($\mathcal{L}_{cluster}$), the Fairness Loss from Class Distribution ($\mathcal{L}_{class}$), and the Conditional Structural Consistency loss ($\mathcal{L}_{cons}$)

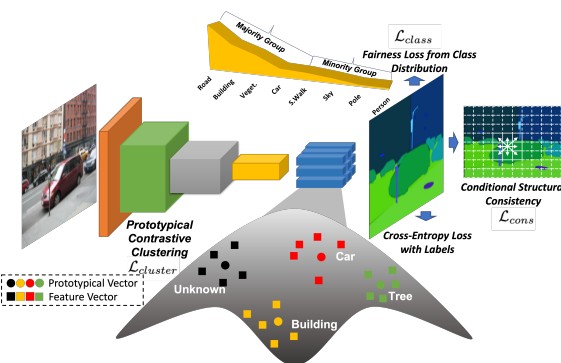

classes forcing the model to behave fairly based on the class distribution. Under the ideal distribution assumption where the model is expected to perform fairly, the $q(y_{i,j})$ will be considered as a uniform distribution $\mathcal{U}$, i.e., $q(y_{i,j}) \sim \mathcal{U}(C)$ ($C$ is the number of classes).

- **The Conditional Structural Consistency Objective** The third term, i.e., $\mathcal{L}_{cons} = \log\left(\frac{q(\mathbf{y}_{\setminus(i,j)}|y_{i,j})}{p(\mathbf{y}_{\setminus(i,j)}|y_{i,j})}\right)$, regularizes the structural consistency of the prediction. This objective acts as a metric to constrain the structure of the predicted segmentation under the ideal distribution assumption. To model this conditional structure consistency, we introduce a conditional structural constraint based on the Markovian assumption discussed in Sec. 3.4.

## 3.2 Prototypical Contrastive Clustering Loss for Handling Unknown Classes

A successful continual learning approach should be able to model background classes without explicit supervision and confront the forgetting problem of previously learned classes when the labels of the task are provided to update the knowledge of the model [20]. A straightforward adoption of Softmax could not be enough to handle. Indeed, the unlabeled pixels will be ignored during training. Thus, it results in these unannotated pixels could be treated as negative samples. Consequently, the segmentation model tends to produce indiscriminative features for these unknown classes that limit the capability of learning new classes in future tasks or recognizing the classes learned previously.

To address this limitation, in addition to the Softmax loss, we introduce a novel Prototypical Contrastive Clustering Loss. In particular, the semantic segmentation pixel belonging to each class can be represented in latent space. Inspired by [20, 2, 21], the features representing the classes can be separated by defining it as a contrastive clustering problem [20] where features of the same class would be pulled closer while features of different classes would be pushed far away. In addition, the deep representations of unknown classes will be grouped into the same cluster of unknown classes to produce discriminative features against other classes.

Formally, for each class $c \in \mathcal{C}^{1..t}$, it is represented by a prototypical vector $\mathbf{p}_c$. In addition, the additional prototypical vector $\mathbf{p}_0$ represents a cluster of unknown classes. Let $f_{i,j}^t$ be a feature representation of pixel at location $(i, j)$ of the input $\mathbf{x}^t$. Then, the Prototypical Contrastive Clustering Loss $\mathcal{L}_{cluster}$ can be defined via a distance $\mathcal{D}$ as follows:

$$\mathcal{L}_{cluster}(\mathbf{x}^t, \mathcal{F}, \theta_t) = \sum_{i,j}\sum_c \mathcal{D}(\mathbf{f}_{i,j}^t, \mathbf{p}_c) \tag{8}$$

$$\mathcal{D}(\mathbf{f}_{i,j}^t, \mathbf{p}_c) = \begin{cases} \ell(\mathbf{f}_{i,j}^t, \mathbf{p}_c) & \text{If } \hat{y}_{i,j}^t = c \\ \max\{0, \Delta - \ell(\mathbf{f}_{i,j}^t, \mathbf{p}_c)\} & \text{otherwise} \end{cases} \tag{9}$$

where $\ell$ is a distance metric, $\Delta$ is a margin between the feature vectors of different classes, and $\hat{y}_{i,j}^t$ is the label of pixel $(i, j)$. Minimizing this loss separates the classes represented in the latent space. For step $t > 1$, $\hat{y}_{i,j}^t$ of an unknown-class pixel will utilize a pseudo label where its assigned label is computed based on the closest cluster. In addition, since the prototypical vectors of classes $c \in \mathcal{C}^{1..t-1}$ have been well learned to represent for classes, these vectors $\mathbf{p}_c$ (where $c \in \mathcal{C}^{1..t-1}$) will be frozen at step $t$ to maintain its learned knowledge of classes $\mathcal{C}^{1..t-1}$.

The set of prototypical vectors at current step $t$, i.e., $\mathbf{p}_0$ and $\mathbf{p}_c$ where $c \in \mathcal{C}^t$ are updated gradually with respect to the growth of feature vectors. In particular, the prototypical vector $\mathbf{p}_c$ will be

periodically updated (after every $M$ iterations) with momentum $\eta$ based on the set of features $\mathbf{f}_{i,j}$ of class $c$. Following common practices [20, 16], to effectively support the updating step and memory efficiency, for each class $c$, we only maintain a set of features $\mathcal{S}_c$ with a fixed length of $L$. Algorithm 1 in the supplementary illustrates an overview of computing the prototypical contrastive clustering loss while updating the class prototypes. Fig. 2 illustrates our proposed FairCL framework.

### 3.3 Prototypical Constrative Clustering Loss to Catastrophic Forgetting

Knowledge Distillation is a common continual learning approach [36, 3, 12, 46] where the knowledge of the previous model will be distilled into the current model. This mechanism prevents the segmentation model from diverging knowledge learned previously and avoiding the catastrophic forgetting problem. This continual learning paradigm has been widely adopted due to its efficiency in computation. In addition, this approach also does not require data rehearsal, i.e., storing the data samples of previous tasks. In this paper, we demonstrate that our Prototypical Constrative Clustering approach is a comprehensive upper limit of the Knowledge Distillation approach. In particular, the common knowledge distillation approach can be formulated as follows:

$$\mathcal{L}_{distill}(\mathbf{x}^t, \mathcal{F}, \theta_t, \theta_{t-1}) = \mathcal{D}(\mathbf{f}^{t-1}, \mathbf{f}^t) \tag{10}$$

where $\mathbf{f}^t$ and $\mathbf{f}^{t-1}$ are the features of the input $\mathbf{x}^t$ produced by the segmentation model at step $t$ and step $t-1$, respectively; and the distance metric $\mathcal{D}$ measure the knowledge gap between $\mathbf{f}^t$ and $\mathbf{f}^{t-1}$.

**Proposition 1:** *The Prototypical Constrative Clustering Approach is the generalized upper bound of the Knowledge Distillation Approach.*

$$\mathcal{L}_{distill}(\mathbf{x}^t, \mathcal{F}, \theta_t, \theta_{t-1}) = \mathcal{O}\left(\mathcal{L}_{cluster}(\mathbf{x}^t, \mathcal{F}, \theta_t)\right) \tag{11}$$

**Proof:** Without lack of generality, we assume that $\mathcal{D}$ is a metric that measures distance between features. Given a set of fixed prototypical vectors $\mathbf{p}_c$, we consider the following triangle inequality of the metric $\mathcal{D}$ as follows:

$$\forall c \in \{0\} \cup \mathcal{C}^{1..t}: \quad \mathcal{D}(\mathbf{f}^t, \mathbf{f}^{t-1}) \leq \mathcal{D}(\mathbf{f}^t, \mathbf{p}_c) + \mathcal{D}(\mathbf{p}_c, \mathbf{f}^{t-1})$$
$$\Leftrightarrow \mathcal{D}(\mathbf{f}^t, \mathbf{f}^{t-1}) \leq \frac{1}{|\mathcal{C}|} \sum_c \left[\mathcal{D}(\mathbf{f}^t, \mathbf{p}_c) + \mathcal{D}(\mathbf{p}_c, \mathbf{f}^{t-1})\right] \tag{12}$$

where $|\mathcal{C}|$ is the number of prototypical vectors. The prototypical vectors $\mathbf{p}_c$ and the feature vectors $\mathbf{f}^{t-1}$ produced by the segmentation model at step $t-1$ are considered as constant features as the model in the previous model $t-1$ has been fixed at step $t$. Thus, the distance $\mathcal{D}(\mathbf{p}_c, \mathbf{f}^{t-1})$ could be considered as constant number. Then, Eqn. (12) can be further derived as follows:

$$\mathcal{D}(\mathbf{f}^t, \mathbf{f}^{t-1}) = \mathcal{O}\left(\frac{1}{|\mathcal{C}|} \sum_c \left[\mathcal{D}(\mathbf{f}^t, \mathbf{p}_c) + \mathcal{D}(\mathbf{p}_c, \mathbf{f}^{t-1})\right]\right) = \mathcal{O}\left(\sum_c \mathcal{D}(\mathbf{f}^t, \mathbf{p}_c)\right) \tag{13}$$
$$\Rightarrow \mathcal{L}_{distill}(\mathbf{x}^t, \mathcal{F}, \theta_t, \theta_{t-1}) = \mathcal{O}\left(\mathcal{L}_{cluster}(\mathbf{x}^t, \mathcal{F}, \theta_t)\right)$$

Intuitively, under this upper bound in Eqn. (13), by only optimizing our prototypical contrastive clustering loss, the knowledge distillation constraint has also been implicitly imposed. Beyond the property of generalized upper bound stated in **Proposition 1**, our approach offers other benefits over the knowledge distillation approach. In particular, our approach is computationally efficient, where our method only requires a single forward pass of the segmentation model. Meanwhile, the knowledge distillation demands two forward passes for both the current and previous models, which also requires additional computational memory for the previous model. Moreover, our approach provides a better representation of each class $c$ through the prototypical vector $\mathbf{p}_c$. This mechanism helps to effectively maintain the knowledge of classes learned previously while allowing the model to update the new knowledge without rehearsing the old data.

### 3.4 Learning Conditional Structural Consistency

The conditional structural constraint plays an important role as it will ensure the consistency of the predicted segmentation map. However, modeling the conditional structural constraint $\log\left(\frac{q(\mathbf{y}_{\setminus(i,j)}|y_{i,j})}{p(\mathbf{y}_{\setminus(i,j)}|y_{i,j})}\right)$ in Eqn. (7) is a quite challenging problem due to two factors, i.e., (1) the unknown ideal conditional

distribution $q(\mathbf{y}_{\setminus(i,j)}|y_{i,j})$, and the complexity of the distribution $q(\mathbf{y}_{\setminus(i,j)}|y_{i,j})$. To address the first limitation of unknown ideal distribution, let us consider the following tight bound as follows:

$$\mathbb{E}_{\mathbf{x}\sim p(\mathbf{x})}\log\left(\frac{q(\mathbf{y}_{\setminus(i,j)}|y_{i,j})}{p(\mathbf{y}_{\setminus(i,j)}|y_{i,j})}\right) \leq -\mathbb{E}_{\mathbf{x}\sim p(\mathbf{x})}\log p(\mathbf{y}_{\setminus(i,j)}|y_{i,j}) \tag{14}$$

The inequality in Eqn. (14) always hold with respect to any form ideal distribution $q(\cdot)$. Thus, optimizing the negative log-likelihood of $\log p(\mathbf{y}_{\setminus(i,j)}|y_{i,j})$ could also regularize the conditional structural constraint due to the upper bound of Eqn. (14). More importantly, *the requirement of ideal data distribution during training has also been relaxed.* However, up to this point, the second limitation of modeling the complex distribution $q(\mathbf{y}_{\setminus(i,j)}|y_{i,j})$ has still not been solved.

To address this problem, we adopt the Markovian assumption [5, 39] to model conditional structural consistency. In particular, we propose a simple yet effective approach to impose the consistency of the segmentation map through the prediction at location $(i, j)$ and predictions of its neighbor pixels. Formally, the conditional structure consistency can be formed via the Gaussian kernel as follows:

$$-\log p(\mathbf{y}_{\setminus(i,j)}|y_{i,j}) \propto \sum_{(i',j')\in\mathcal{N}(i,j)} \exp\left\{-\frac{||x_t^{i,j}-x_{i',j'}^t||_2^2}{2\sigma_1^2} - \frac{||y_t^{i,j}-y_{i',j'}^t||_2^2}{2\sigma_2^2}\right\} \tag{15}$$

where $\mathcal{N}(i,j)$ is the set of neighbor pixels of $(i,j)$, $\{\sigma_1, \sigma_2\}$ are the scale hyper-parameters of the Gaussian kernels. The conditional structural consistency loss defined in Eqn. (15) enhance the smoothness and maintain the consistency of the predicted segmentation map by imposing similar predictions of neighbor pixels with similar contextual colors.

## 4 Experiments

In this section, we first describe the datasets and metrics used in our experiments. Then, we present the ablation studies to illustrate the effectiveness of our proposed method. Finally, we compare our approach with prior CSS methods to demonstrate our SOTA performance.

### 4.1 Datasets and Evaluation Protocols

**Datasets** *ADE20K* [49] is a semantic segmentation dataset that consists of more than 20K scene images of 150 semantic categories. Each image has been densely annotated with pix-level objects and objects parts labels. *Cityscapes* [10] is a real-world autonomous driving dataset collected in European. This dataset includes $3,975$ urban images with high-quality, dense labels of 30 semantic categories. *PASCAL VOC* [13] is a common dataset that consists of more than 10K images of 20 classes.

**Implementation** Two segmentation network architectures are used in our experiments, i.e., (1) DeepLab-V3 [6] with the ResNet-101 backbone, and (2) SegFormer [45] with MiT-B3 backbone. Further details of our implementation will be available in our supplementary.

**Evaluation Protocols:** Following [12], we focus on the overlapped CSS evaluation. Our proposed method is evaluated on several settings for each dataset, i.e., ADA20K 100-50 (2 steps), ADA20K 100-10 (6 steps), ADA20K 100-5 (11 steps), Cityscapes 11-5 (3 steps), Cityscapes 11-1 (11 steps),

| Backbone | $\mathcal{L}_{cluster}$ | $\mathcal{L}_{class}$ | $\mathcal{L}_{cons}$ | 0-100 | | 100-150 | | all | | avg | |
|---|---|---|---|---|---|---|---|---|---|---|---|
| | | | | mIoU | STD | mIoU | STD | mIoU | STD | mIoU | STD |
| DeepLab-V3 | | | | 0.08 | 0.84 | 19.52 | 20.18 | 6.52 | 13.14 | 24.41 | 13.14 |
| | ✓ | | | 41.71 | 19.90 | 15.33 | 21.96 | 32.97 | 23.03 | 37.58 | 23.03 |
| | ✓ | ✓ | | 42.25 | 19.31 | 18.55 | 20.52 | 34.40 | 22.07 | 38.35 | 22.07 |
| | ✓ | ✓ | ✓ | **43.40** | **19.08** | **24.04** | **19.12** | **36.99** | **21.67** | **40.45** | **21.67** |
| SegFormer | | | | 0.10 | 0.84 | 23.18 | 19.83 | 7.74 | 15.74 | 25.82 | 15.74 |
| | ✓ | | | 43.40 | 19.35 | 21.60 | 22.06 | 36.18 | 22.32 | 39.85 | 22.32 |
| | ✓ | ✓ | | 43.35 | 19.03 | 23.50 | 20.75 | 36.78 | 21.86 | 40.34 | 21.86 |
| | ✓ | ✓ | ✓ | **43.56** | **18.71** | **25.46** | **19.99** | **37.56** | **21.10** | **40.73** | **21.10** |

Table 1: Effectiveness of our approach on the ADE20K 100-50 benchmark. Two different networks, i.e., DeepLab-V2 [5] and SegFormer [45], and three different losses, i.e., Prototypical Contrastive Clustering ($\mathcal{L}_{cluster}$), Fairness Loss ($\mathcal{L}_{class}$), and Conditional Structural Consistency ($\mathcal{L}_{cons}$).

Cityscapes 1-1 (21 steps), Pascal VOC 15-1 (3 steps), and Pascal VOC 10-1 (11 steps). The mean Intersection over Union (mIoU) metric is used in our experiments. The mIoU is computed after the last step for the classes learned from the first step, the later continual classes, and all classes. The mIoU for the initial classes shows the robustness of the model to catastrophic forgetting, while the metric for the later classes reflects the ability to learn new classes. To measure the fairness of the model, we also report the standard deviation (STD) of IoUs over all classes.

## 4.2 Ablation Study

Our ablative experiments study the effectiveness of our proposed FairCL approach on the performance of the CSS model and fairness improvement on the ADE20K 100-50 benchmark (Table 1).

**Effectiveness of the Network Backbone** Table 1 illustrates the results of our approach using the DeepLab-V3 [5] with the Resnet101 backbone and the SegFormer [45] with a Transformer backbone, i.e. MiT-B3 [45]. As shown in our results, the performance of segmentation models using a more powerful backbone, i.e., Transformer, outperforms the models using the Resnet backbone. The capability of learning new classes has been improved notably, i.e., the mIoU of classes 101-150 in the full configuration has been improved from 19.86% to 25.46% while the model keeps robust to catastrophic forgetting, the mIoU has been increased from 41.96% to 43.56% in the classes 0-100. Additionally, fairness between classes has been promoted when the standard deviation of the IoU over classes has been reduced from 21.67% to 21.10%.

| Method | 11-5 3 steps | 11-1 11 steps | 1-1 21 steps |
|---|---|---|---|
| Joint | 79.30 | 79.30 | 79.30 |
| LWF-MC [31] | 58.90 | 56.92 | 31.24 |
| ILT [25] | 59.14 | 57.75 | 30.11 |
| MiB [3] | 61.51 | 60.02 | 42.15 |
| PLOP [12] | 63.51 | 62.05 | 45.24 |
| RCIL [46] | 64.30 | 63.00 | 48.90 |
| FairCL + DeepLab-V3 | 66.96 | 66.61 | 49.22 |
| FairCL + SegFormer | **67.85** | **67.09** | **55.68** |

Table 2: Final mIoU (%) for Continual Semantic Segmentation on Cityscapes.

**Effectiveness of the Prototypical Contrastive Clustering Loss** We evaluate the impact of the Prototypical Contrastive Clustering Loss ($\mathcal{L}_{cluster}$) in improving the performance in the continual learning problem compared to the fine-tuning approach. As shown in Table 1, the clustering loss has significant improvements in the catastrophic forgetting robustness compared to using only the Softmax loss. In particular, the mIoU of classes 0-100 for both DeepLab-V3 and SegFormer backbones has been improved by $+41.63\%$ and $+43.30\%$ respectively that makes the overall mIoU increase by $+26.45\%$ and $+28.44\%$, and the average mIoU between classes increases by $+13.17\%$ and $+14.03\%$. Although the STD of IoUs has slightly increased in this setting, the major target of our $\mathcal{L}_{cluster}$ is used to model the catastrophic forgetting and background shift problems in CSS illustrated by the significant performance improvement of mIoU.

**Effectiveness of the Fairness Treatment Loss** As reported in Table 1, the fairness treatment from the class distribution loss $\mathcal{L}_{class}$ significantly improves the overall performance and the accuracy of classes. In detail, the STD of IoU from classes has been reduced by $0.96\%$ and $0.46\%$ for both backbones while the mIoU has been improved from $32.97\%$ to $34.40\%$ and from $36.18\%$ to $36.78\%$, respectively. The results have shown that our approach has promoted the fairness of the model.

**Effectiveness of the Conditional Structural Consistency** The full configuration in Table 1 shows experimental results of our model using conditional structure constraint loss $\mathcal{L}_{cons}$. As illustrated in our results, the conditional structure constraint demonstrates effective improvement. Indeed, it

| Method | 100-50 (2 steps) | | | | 50-50 (3 steps) | | | | 100-10 (6 steps) | | | |
|---|---|---|---|---|---|---|---|---|---|---|---|---|
| | 0-100 | 101-150 | *all* | *avg* | 0-50 | 51-150 | *all* | *avg* | 0-100 | 101-150 | *all* | *avg* |
| Joint | 44.30 | 28.20 | 38.90 | - | 51.10 | 32.80 | 38.90 | - | 44.30 | 28.20 | 38.90 | - |
| ILT [25] | 18.29 | 14.40 | 17.00 | 29.42 | 3.53 | 12.85 | 9.70 | 30.12 | 0.11 | 3.06 | 1.09 | 12.56 |
| MiB [3] | 40.52 | 17.17 | 32.79 | 37.31 | 45.57 | 21.01 | 29.31 | 38.98 | 38.21 | 11.12 | 29.24 | 35.12 |
| PLOP [12] | 41.87 | 14.89 | 32.94 | 37.39 | 48.83 | 20.99 | 30.40 | 39.42 | 40.48 | 13.61 | 31.59 | 36.64 |
| RCIL [46] | 42.30 | 18.80 | 34.50 | 38.48 | 48.30 | 25.00 | 32.50 | - | 39.30 | 17.60 | 32.10 | - |
| MiB + AWT [15] | 40.90 | 24.70 | 35.60 | - | 46.60 | 26.85 | 33.50 | - | 39.10 | 21.28 | 33.20 | - |
| SSUL [4] | 41.28 | 18.02 | 33.58 | - | 48.38 | 20.15 | 29.56 | - | 40.20 | 18.75 | 33.10 | - |
| SATS [30] | - | - | - | - | - | - | - | - | 41.42 | 19.09 | 34.18 | - |
| FairCL + DeepLab-V3 | 43.40 | 24.04 | 36.99 | 40.45 | **49.65** | 26.84 | 34.55 | 41.68 | 41.73 | 20.36 | 34.65 | 39.01 |
| FairCL + SegFormer | **43.56** | **25.46** | **37.56** | **40.73** | 49.62 | **27.78** | **35.15** | **42.25** | **42.21** | **21.91** | **35.49** | **39.36** |

Table 3: Continual Semantic Segmentation results on ADE20k in Mean IoU (%).

promotes the accuracy of the initial classes and the novel classes when the mIoU has been increased from 43.35% to 43.56% and from 23.50% to 25.46% respectively with the Transformer backbone. The fairness of classes is also improved as the standard deviation of the IoU of classes 0-100 and classes 101-150 is reduced from 19.03% to 18.71% and from 20.75% to 19.99%.

### 4.3 Comparison with State-of-the-Art Methods

**Cityscapes** As shown in Table 2, our FairCL outperforms previous SOTA methods evaluated on Cityscapes benchmarks. In particular, in the 11-5 task, our method using Resnet and Transformer achieves the mIoU of 66.96% and 67.85% respectively which shows better performance than prior methods. Meanwhile, the results for the 11-1 task are 66.61% and 67.09% w.r.t. the Resnet and Transformer backbones. For the 1-1 task, the mIoU of our method is 49.22% and 55.68%.

**ADE20K** Table 3 presents our experimental results using ResNet and Transformer backbones compared to prior SOTA approaches. Our proposed approach achieves SOTA performance and outperforms prior methods. In particular, our approach achieves the final mIoU of 36.99% for Resnet and 37.56% for Transformer in the 100-

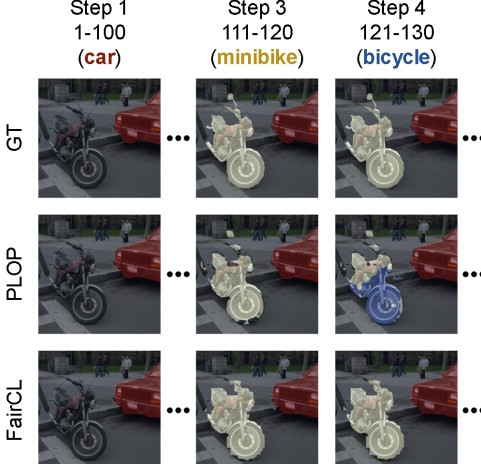

Figure 3: Qualitative results of Our Approach and PLOP [12] on ADE20K 100-10.

50 tasks. For the 50-50 tasks, the model reaches the final mIoU of 34.55% and 35.15% for the Resnet and Transformer backbones, respectively while the result of the prior method [15] is 33.50%. Meanwhile, the overall results of our method for the 100-10 task are 34.65% and 35.49% which shows outperforming previous approaches. Fig. 3 visualizes the qualitative result of our method compared to PLOP [12]. Initially, the ground truth contains the class "car" in the first step and the class "minibike" in the third step. Then, in the fourth step, the class "bicycle" is included. As a result, PLOP [12] partly forgets the "minibike" information when learning the class "bicycle" information. Meanwhile, our method consistently maintains the information of "minibike" and predicts segmentation correctly.

| Method | 15-1 (6 steps) | | | 10-1 (11 steps) | | |
|---|---|---|---|---|---|---|
| | 0-15 | 16-20 | all | 0-10 | 11-20 | all |
| Joint | 79.8 | 72.4 | 77.4 | 78.4 | 76.4 | 77.4 |
| LWF [31] | 6.0 | 3.9 | 5.5 | 8.0 | 2.0 | 4.8 |
| ILT [25] | 9.6 | 7.8 | 9.2 | 7.2 | 3.7 | 5.5 |
| MiB [3] | 38.0 | 13.5 | 32.2 | 20.0 | 20.1 | 20.1 |
| SDR [26] | 47.3 | 14.7 | 39.5 | 32.4 | 17.1 | 25.1 |
| PLOP [12] | 65.1 | 21.1 | 54.6 | 44.0 | 15.5 | 30.5 |
| RCIL [46] | 70.6 | **23.7** | 59.4 | 55.4 | 15.1 | 34.3 |
| FairCL + DeepLab-V3 | 72.0 | 22.7 | 60.3 | 42.3 | **25.6** | 34.4 |
| FairCL + SegFormer | **73.5** | 22.8 | **61.5** | **57.1** | 14.2 | **36.6** |

Table 4: The mIoU (%) of CSS on Pascal VOC.

**Pascal VOC** As shown in Table 4, the proposed method outperforms the prior approaches evaluated on the Pascal VOC 2012 dataset. In detail, our method achieves the overall mIoU of 61.5% in the 15-1 task while the result of the previous method [46] is 59.4%. Meanwhile, the mIoU in the 10-1 task is 36.6% which shows better performance than the prior methods.

## 5 Conclusions and Dicussions

**Conclusions:** This paper introduces a novel fairness continual learning approach in semantic segmentation by analyzing the effect of class distribution. Furthermore, a new learning paradigm of continual learning, i.e., the prototypical Contrastive clustering loss, is proposed to sufficiently address the catastrophic forgetting and the background shift problems. The experimental results on the three CSS benchmarks, i.e., ADE20K, Cityscapes, and Pascal VOC, have shown our SOTA performance.

**Limitations:** Our paper has chosen specific configurations of network backbones and hyper-parameters to support our hypothesis. However, the other aspects of learning have not been fully investigated, e.g., learning hyper-parameters, the selected neighbor pixels, or other forms of $\mathcal{L}_{cons}$.

**Broader Impacts:** This work studies the problem of Fairness in Continual Learning which is a step toward fairness awareness in continual semantic segmentation. Our contributions emphasize the importance of fairness in continual semantic segmentation learning and provide a solution to address the fairness concern that increases the robustness and trustworthiness of the segmentation model.

**Acknowledgment** This work is partly supported by NSF Data Science, Data Analytics that are Robust and Trusted (DART), and Googler Initiated Research Grant. We also thank Utsav Prabhu for invaluable discussions and suggestions and acknowledge the Arkansas High-Performance Computing Center for providing GPUs.

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
