# Supplementary
# Fairness Continual Learning Approach to Semantic Scene Understanding in Open-World Environments

**Thanh-Dat Truong[1], Hoang-Quan Nguyen[1], Bhiksha Raj[2,3], and Khoa Luu[1]**
[1]CVIU Lab, University of Arkansas, Fayetteville, AR, 72701
[2]Carnegie Mellon University, Pittsburgh, PA, 15213
[3]Mohammed bin Zayed University of AI, Abu Dhabi, UAE
{tt032, hn016, khoaluu}@uark.edu, bhiksha@cs.cmu.edu

## 1 Proof of Eqn. (6)

$$
\begin{aligned}
\theta^* &= \int \mathcal{L}(\mathbf{y}, \hat{\mathbf{y}}) q(\mathbf{y}) q(\hat{\mathbf{y}}) d\mathbf{y} d\hat{\mathbf{y}} \\
&= \int \mathcal{L}(\mathbf{y}, \hat{\mathbf{y}}) \frac{q(\mathbf{y})}{p(\mathbf{y})} \frac{q(\hat{\mathbf{y}})}{p(\hat{\mathbf{y}})} p(\mathbf{y}) p(\hat{\mathbf{y}}) d\mathbf{y} d\hat{\mathbf{y}}
\end{aligned}
\tag{1}
$$

It should be noted that the fraction $\frac{q(\hat{\mathbf{y}})}{p(\hat{\mathbf{y}})}$ could be considered as constants as $q(\hat{\mathbf{y}})$ and $p(\hat{\mathbf{y}})$ are distriuted over the ground-truth segmentation. Thus, it would be ignored during the optimization process. Then, the formula can be further derived as follows:

$$
\begin{aligned}
\theta^* &\simeq \int \mathcal{L}(\mathbf{y}, \hat{\mathbf{y}}) \frac{q(\mathbf{y})}{p(\mathbf{y})} p(\mathbf{y}) p(\hat{\mathbf{y}}) d\mathbf{y} d\hat{\mathbf{y}} \\
&= \arg\min_{\theta} \left[ \mathbb{E}_{\mathbf{x} \sim p(\mathbf{y}), \hat{\mathbf{y}} \sim p(\hat{\mathbf{y}})} \mathcal{L}(\mathbf{y}, \hat{\mathbf{y}}) \frac{q(\mathbf{y})}{p(\mathbf{y})} \right] \\
&= \arg\min_{\theta} \left[ \mathbb{E}_{\mathbf{x} \sim p(\mathbf{y}), \hat{\mathbf{y}} \sim p(\hat{\mathbf{y}})} \sum_{i,j} \mathcal{L}(y_{i,j}, \hat{y}_{i,j}) \frac{q(y_{i,j}) q(\mathbf{y}_{\backslash(i,j)}|y_{i,j})}{p(y_{i,j}) p(\mathbf{y}_{\backslash(i,j)}|y_{i,j})} \right]
\end{aligned}
\tag{2}
$$

## 2 Proof of Eqn. (7)

By taking the logarithm, the optimization process can be rewritten as follows:

$$
\begin{aligned}
\theta^* &= \arg\min_{\theta} \mathbb{E}_{\mathbf{x} \sim p(\mathbf{x}), \hat{\mathbf{y}} \sim p(\hat{\mathbf{y}})} \mathcal{L}(\mathbf{y}, \hat{\mathbf{y}}) \frac{q(\mathbf{y})}{p(\mathbf{y})} \\
&\simeq \arg\min_{\theta} \mathbb{E}_{\mathbf{x} \sim p(\mathbf{x}), \hat{\mathbf{y}} \sim p(\hat{\mathbf{y}})} \log\left( \mathcal{L}(\mathbf{y}, \hat{\mathbf{y}}) \frac{q(\mathbf{y})}{p(\mathbf{y})} \right) \\
&= \arg\min_{\theta} \mathbb{E}_{\mathbf{x} \sim p(\mathbf{x}), \hat{\mathbf{y}} \sim p(\hat{\mathbf{y}})} \left( \log \mathcal{L}(\mathbf{y}, \hat{\mathbf{y}}) + \log \frac{q(\mathbf{y})}{p(\mathbf{y})} \right) \\
&= \arg\min_{\theta} \mathbb{E}_{\mathbf{x} \sim p(\mathbf{x}), \hat{\mathbf{y}} \sim p(\hat{\mathbf{y}})} \left[ \log \mathcal{L}(\mathbf{y}, \hat{\mathbf{y}}) + \frac{1}{N} \sum_{i,j} \log\left( \frac{q(y_{i,j}) q(\mathbf{y}_{\backslash(i,j)}|y_{i,j})}{p(y_{i,j}) p(\mathbf{y}_{\backslash(i,j)}|y_{i,j})} \right) \right] \\
&= \arg\min_{\theta} \mathbb{E}_{\mathbf{x} \sim p(\mathbf{x}), \hat{\mathbf{y}} \sim p(\hat{\mathbf{y}})} \left\{ \log \mathcal{L}(\mathbf{y}, \hat{\mathbf{y}}) + \frac{1}{N} \sum_{i,j} \left[ \log\left( \frac{q(y_{i,j})}{p(y_{i,j})} \right) + \log\left( \frac{q(\mathbf{y}_{\backslash(i,j)}|y_{i,j})}{p(\mathbf{y}_{\backslash(i,j)}|y_{i,j})} \right) \right] \right\}
\end{aligned}
$$

37th Conference on Neural Information Processing Systems (NeurIPS 2023).

where $N$ is the total number of pixels. In addition, minimizing $\log \mathcal{L}(\mathbf{y}, \hat{\mathbf{y}})$ is equivalent to minimizing $\mathcal{L}(\mathbf{y}, \hat{\mathbf{y}})$. Therefore, the formula can be further derived as follows:

$$\theta^* = \arg\min_{\theta} \mathbb{E}_{\mathbf{x} \sim p(\mathbf{x}), \hat{\mathbf{y}} \sim p(\hat{\mathbf{y}})} \left\{ \mathcal{L}(\mathbf{y}, \hat{\mathbf{y}}) + \frac{1}{N} \sum_{i,j} \left[ \log\left(\frac{q(y_{i,j})}{p(y_{i,j})}\right) + \log\left(\frac{q(\mathbf{y}_{\setminus(i,j)}|y_{i,j})}{p(\mathbf{y}_{\setminus(i,j)}|y_{i,j})}\right) \right] \right\}$$

## 3 Protypical Contrastive Clustering Algorithm

Inspired by [7, 6], we develop the algorithm to compute the Prototypical Contrastive Clustering loss and update prototypical vectors. Algorithm 1 illustrates the procedure to compute the Prototypical Contrastive Clustering loss and update the prototypical vectors.

---

Algorithm 1: Prototypical Constrative Clustering Loss

---

**Input:** Current iteration $i$ of step $t$; A set of prototypical vectors $\{p_c\}_{c=0}^{|\mathcal{C}^{1..t}|}$; A set of features $\mathbf{f}_{i,j}$; Momentum
   parameter: $\eta$; A set of storing features $\{\mathcal{S}_c\}_{c=0}^{|\mathcal{C}^{1..t}|}$
1: Initialize $\mathbf{p}_c$ where $c \in \mathcal{C}^t$ in the first iteration.
2: $\mathcal{L}_{cluster} \leftarrow 0$
3: **if** $i == M$ **then**
4:     **For each** $c \in \{0\} \cup \mathcal{C}^t$
5:         $\mathbf{p_c} \leftarrow \mathbb{E}_{\mathbf{f} \in \mathcal{S}_c} \mathbf{f}$.
6:     $\mathcal{L}_{cluster} \leftarrow$ Compute Prototypical Constrative Clustering Loss based on Eqn. (9).
7: **else if** $i > M$ **then**
8:     **if** $i\%M == 0$ **then**
9:         **For each** $c \in \{0\} \cup \mathcal{C}^t$
10:            $\mathbf{p_c} \leftarrow \eta\mathbf{p}_c + (1 - \eta)\mathbb{E}_{\mathbf{f} \in \mathcal{S}_c} \mathbf{f}$.
11:    $\mathcal{L}_{cluster} \leftarrow$ Compute Prototypical Constrative Clustering Loss based on Eqn. (9).
12: **return** $\mathcal{L}_{cluster}$

---

## 4 Implementation

Two segmentation network architectures have been used in our experiments, i.e., (1) DeepLab-V3 [2] with the ResNet-101 backbone, and (2) SegFormer [10] with MiT-B3 backbone. Our framework is implemented in PyTorch and trained on four 40GB-VRAM NVIDIA A100 GPUs. The model is optimized by the SGD optimizer [1] with momentum 0.9, weight decay $10^{-4}$, and batch size of 6 per GPU. The learning rate is set individually for each step and dataset. In particular, the learning rate for the initial step and the continual steps of the ADE20K dataset is $10^{-2}$ and $10^{-3}$ respectively, while the learning rate for the Cityscapes experiment is $2 \times 10^{-2}$ and $2 \times 10^{-3}$. The feature vectors from the last layer of the decoder are used for the prototypical clustering loss. For each class, the number of feature vectors in each set $\mathcal{S}_c$ for computing the prototypes is 500 features. Following common practices in contrastive learning [7, 6], we adopt the Euclidean distance for our $\ell$ in the Prototypical Contrastive Clustering loss $\mathcal{L}_{cluster}$ and the margin $\nabla$ between features of different classes is set to 10. The momentum $\eta$ to update the prototypical vectors is set to 0.99. Following [2, 9], in the conditional structural consistency loss, the number of neighbor pixels is within a window size of $3 \times 3$.

## 5 Additional Experiments

### 5.1 Performance Improvement of Major and Minor Groups

To illustrate the performance improvement of our proposed method in major and minor classes, we include the results of the mIoU (all) and the STD among IoUs of the major group and the minor group on the ADE20K 100-50 (Table 1) and Cityscapes 11-5 (Table 2) benchmarks. As shown in the table below, our proposed approach has improved the performance of both major and minor groups. Thus, these results illustrated that the performance improvement in mIoU is also coming from the minority classes. It helps to enhance the mIoU performance and reduce the STD in both major and minor classes, thus, improving the fairness of the model predictions.

| Backbone | $\mathcal{L}_{cluster}$ | $\mathcal{L}_{class}$ | $\mathcal{L}_{cons}$ | Major Group mIoU | Major Group STD | Minor Group mIoU | Minor Group STD |
|---|---|---|---|---|---|---|---|
| | ✓ | | | 48.78 | 18.12 | 25.13 | 21.12 |
| DeepLab-V3 | ✓ | ✓ | | 48.89 | 17.87 | 27.24 | 20.76 |
| | ✓ | ✓ | ✓ | 50.11 | 17.46 | 30.52 | 20.43 |

Table 1: ADE20K 150-50 Benchmark

| Backbone | $\mathcal{L}_{cluster}$ | $\mathcal{L}_{class}$ | $\mathcal{L}_{cons}$ | Major Group mIoU | Major Group STD | Minor Group mIoU | Minor Group STD |
|---|---|---|---|---|---|---|---|
| | ✓ | | | 87.44 | 9.25 | 53.37 | 16.72 |
| DeepLab-V3 | ✓ | ✓ | | 88.29 | 8.85 | 55.72 | 13.39 |
| | ✓ | ✓ | ✓ | 89.20 | 8.41 | 56.70 | 11.96 |

Table 2: Cityscapes 11-5 Benchmark

Similarly, to illustrate the effectiveness and robustness of our method in the non-incremental setting. We report our results after the first learning step on the ADE20K 100-50 (Table 3) and Cityscapes 11-5 (Table 4) benchmarks. Our proposed fairness approach has also contributed to the performance improvement of both major and minor classes in non-incremental settings. The comparison table of major and minor groups in the first step is illustrated below.

| Backbone | $\mathcal{L}_{cluster}$ | $\mathcal{L}_{class}$ | $\mathcal{L}_{cons}$ | Major Group mIoU | Major Group STD | Minor Group mIoU | Minor Group STD |
|---|---|---|---|---|---|---|---|
| | ✓ | | | 49.07 | 18.35 | 35.45 | 19.08 |
| DeepLab-V3 | ✓ | ✓ | | 49.17 | 18.31 | 35.57 | 18.24 |
| | ✓ | ✓ | ✓ | 50.04 | 17.92 | 37.88 | 18.19 |

Table 3: ADE20K 150-50 Benchmark (Non-incremental Setting)

| Backbone | $\mathcal{L}_{cluster}$ | $\mathcal{L}_{class}$ | $\mathcal{L}_{cons}$ | Major Group mIoU | Major Group STD | Minor Group mIoU | Minor Group STD |
|---|---|---|---|---|---|---|---|
| | ✓ | | | 87.64 | 8.96 | 53.78 | 17.16 |
| DeepLab-V3 | ✓ | ✓ | | 88.56 | 8.70 | 56.45 | 13.46 |
| | ✓ | ✓ | ✓ | 89.52 | 8.08 | 57.87 | 12.08 |

Table 4: Cityscapes 11-5 Benchmark (Non-incremental Setting)

## 5.2 The Role of Conditional Structural Consistency

The goal of conditional structural consistency is to improve the prediction gap among neighbor pixels, thus, enhancing the smoothness of the predictions. In addition, it helps to increase fairness among classes. It is because this loss helps to clean up the spurious or ambiguous predictions produced by the major classes around minor classes. Therefore, the loss alleviates the dominance of the major groups and improves the accuracy of the minor groups, thus, resulting in fairness that has also been further improved (as illustrated in Table 1 in the main paper).

## 5.3 The Choice of Margin $\Delta$

We also perform an additional ablation study on the ADE20K (100-50) benchmark to investigate the impact of the delta. As shown in Table 5, the impact of $\Delta$ does not significantly influence the results due to the minor performance drop.

| $\Delta$ | 0-100 | 101-150 | all | avg |
|---|---|---|---|---|
| $\Delta = 5$ | 43.04 | 23.84 | 36.68 | 40.23 |
| $\Delta = 10$ | **43.40** | **24.04** | **36.99** | **40.45** |
| $\Delta = 15$ | 42.89 | 23.92 | 36.61 | 40.26 |

Table 5: The effectiveness of $\Delta$

## 5.4 The Performance Analysis of ADE20K, Cityscapes, and Pascal VOC

We have observed the performance improvement of our approach on Pascal VOC is less significant than the ADE20K and Cityscapes because of the minor bias in Pascal VOC. In particular, the data distributions of these datasets are visualized in Figure 1 in the rebuttal file (the PDF file of the rebuttal is attached in Global Response). We calculated the entropy value of the data distributions that illustrate the balance level of the datasets (the higher value of entropy, the more balance the dataset as the data distribution tends to be more uniform). Then, based on the data distributions and entropy values, we observe the data distributions of ADE20K and Cityscapes suffer more bias than the Pascal VOC since the entropy value of Pascal VOC (H = 0.81) is higher than ADE20K (H = 0.69) and Cityscapes (H = 0.62). Thus, ADE20K and Cityscapes suffer severe fairness issues compared to Pascal VOC. Our approach aims to improve the fairness of the model. Therefore, on the more severe bias datasets (ADE20K and Cityscapes), our approach performs more significantly.

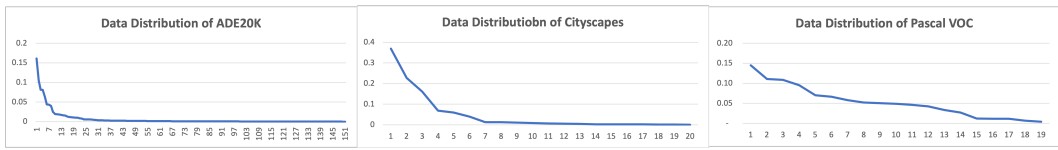

Figure 1: Data Distribution of ADE20K, Cityscapes, and Pascal VOC. The data distributions of ADE20K and Cityscapes suffer a more severe imbalance compared to Pascal VOC.

## 5.5 Memory Efficiency

We would like to highlight that storing prototypical vectors requires significantly less memory than using the additional teacher model as used in distillation approaches [5]. For example, in the ADE20K benchmark, storing DeepLab-V3 (151 classes) requires $58.664M$ parameters, while storing 152 prototypical 2048-D vectors (including the unknown cluster) only uses $0.311M$ parameters. In addition, the computation of loss is cheaper than a forward pass of the entire network used in distillation. Therefore, in terms of computational cost and memory, our approach remains more efficient compared to knowledge distillation approaches.

## 6 Discussion of Limitations

In our paper, we have specified a set of learning hyper-parameters and network designs to support our hypothesis. However, the impact of these learning hyper-parameters on fairness should be investigated. In particular, as aforementioned, the balanced weights among losses should be studied in future work to analyze the impact of each module when these are combined. In our experiments, we have used ResNet 101 and MiB-B3 as the backbone of our segmentation networks. The subsequent work should investigate the effectiveness of different networks, e.g., Swin [8], SegFormer [10], Mask2Former [4, 3], to fairness in continual semantic segmentation. Also, the training batch size, learning schedule, and optimizer could affect the fairness of the model and should be investigated in future research. In addition, the different forms of conditional structure modeling (i.e., $\mathcal{L}_{cons}$ [2, 9] or the number of neighbors pixels in the Markovian formula could impact the fairness predictions of the segmentation model. Future studies should conduct a deep analysis of different forms of conditional structural modeling $\mathcal{L}_{cons}$. These potential limitations will motivate future research to continue improving the contrastive learning approach to fairness continual learning in semantic segmentation.