# OpenReview forum: "Fairness Continual Learning Approach to Semantic Scene Understanding in Open-World Environments"
_NeurIPS.cc/2023/Conference — NeurIPS 2023 poster_

### Official Review · Reviewer_9fXu · 2023-06-27

**Soundness:** 3 good
**Presentation:** 3 good
**Contribution:** 3 good
**Rating:** 6
**Confidence:** 5

**Summary:**

This paper proposes a fairness continual learning approach in semantic segmentation. Derived from the therotical analysis of the effect of class distribution, a new learning paradigm that is consisted of three main objectives: the task-specific objective to handle the catastrophic forgetting problem, the fairness objective that maintains the fairness predictions over classes, and the conditional structural constraint that imposes the consistency of the segmentation predictions. New SOTA results are attained on the standard benchmarks.

**Strengths:**

1. Deriving the learning objective through analysis of the effect of class distribution is novel.
2. Strong performances are  achieved on several standard benchmarks.

**Weaknesses:**

- The title is misleading. In my understanding, the open-world environments means lacking of labels. While the proposed method can hardly work in unsupervised setting.
- Some explanation through mathmatical derivation is not rigorous such as eq. 5 and eq. 12 and 13. More explanation either in the main paper or supplimentary material needed. Detail illustration can be found in the questions.
- Some details are not clear. How the fairness objective $L_{class}$ is calculated during training is not clear and lacking a final integrated training objective. Whether there is hyperparameters between losses is not mentioned.
- According to the ablation study, fairness improvement brought by the fairness objective is minor if measured in STD of IoU over all classes. Better measurement of the fairness is needed. For example, variation of the performance on rare classes and dominant classes with and without fairness loss can be compared in a longer learning steps.


**Questions:**

1. Eq. 5 is hard to understand and I think it could not be intuitively achieved.
2. The explanation that the prototypical contrastive clustering loss is an upper bound of the knowledge distillation approach is not rigorous. In eq. (12) , the $D(p_c, f^{t-1})$ cannot directly ignored as a constant number. When $c$ is a new class from $C^t$, the $p_c$ is changing during learning and $D(p_c, f^{t-1})$ will change. Therefore, $O(D(f^t, p_c)$ is not a tight boundary. Minimizing $L_{cluster}$ is not strictly equals to the minimizing of $L_distill$. From another perspective, although prototypical constrative clustering loss don't need two forward pass of the segmentation model, it have to store the prototypical vector $p_c$ using additional memory. How to calculate the $p_c$ is also critical in achieving good performance.
3. What is the final integrated training loss? Is there any balance weights between different losses?
4. What is the result in the first line of DeepLab-V2 and SegFormer in Table 1?
5. Im my understanding, the conditional structural consistency mitigate the prediction gap among neighbour pixels. However, it will also smooth the edge pixels of an instance. According to the ablation study, it also improves the fairness of classes. Could the author give more explanation on it?
6. Minor problems: when writting the related works, it better to use the surname other than the name. For example, Ziwei et al. should be Liu et al in line 93. In eq. 5, the $y_{i,h}$ should be $y_{i,j}$ in the left term?

**Limitations:**

Authors have pointed out the limitations of the proposed method and some possible improvements.

---

> ### Author Rebuttal · Authors · 2023-08-06
>
> ## Response to Reviewer 9fXu
>
>
>
>
> Dear Reviewer 9fXu,
>
>
>
>
> We thank the reviewer for your careful read and valuable feedback.
> We are pleased you encourage that our theoretical analysis is novel and our approach achieves strong performance. We appreciate your constructive comments and would like to address these points as follows:
>
>
>
>
> [Q1] **Misleading Title**
>
>
>
>
> [A1] We thank you for the comment from the reviewer. However, we would like to clarify our paper title. Basically, continual learning aims to learn the AI model sequentially and address the challenges, e.g., catastrophic forgetting, background shift, and fairness, posed by the evolving and dynamic environments where the new data and classes continuously grow over time. Meanwhile, in the open-world problem setting, the data distribution can change, e.g., data distribution grows, new classes appear, or data of existing classes can evolve. Thus, there is an interference between continual learning and open-world learning in which continual learning could be considered as a part of the goals of open-world learning. Therefore, we use the open-world term in our paper title.
>
>
>
>
> [Q2] **Explanation of Eqn. (5)**
>
>
>
>
> [A2] Intuitively, Eqn (5) illustrates that the major classes of more samples will produce larger gradients to update the classifier and the network backbone. Meanwhile, the minor classes of fewer samples will produce smaller gradients. Thus, the major classes have a higher lower bound of loss value and contribute much higher importance than the minor classes having fewer samples. Therefore, this bias between major and minor classes will influence the unfair predictions produced by the model. A similar issue has been observed in [1, 2, 3].
>
>
>
>
> [Q3] **The upper proof of prototypical contrastive clustering loss**
>
>
>
>
> [A3] We clarify our proof of Eqns (12-13) in Sec. 3.3 still holds true. We acknowledge that $\mathbf{p}_c$ is updated during the training procedure. However, at the point of the prototypical contrastive clustering loss is computed, the set of prototypical vectors $\mathbf{p}_c$ at this point is fixed. Only after each $M$ iterations, the prototypical vectors $\mathbf{p}_c$ will be updated. Thus, our clustering loss remains an upper bound of distillation loss due to the fixed vector $\mathbf{p}_c$ (could be considered as constant) at the loss computation time. Therefore, without a strict argument, it remains that our derive in Eqn (12) still holds. In addition, although our approach does not directly minimize the knowledge distillation loss due to the upper bound, minimizing our clustering loss will intrinsically decrease the upper bound of the knowledge distillation loss. Thus, our approach could also implicitly minimize the knowledge distillation loss and prevent catastrophic forgetting.
>
>
>
>
> [Q4] **Memory efficiency of prototypical contrastive clustering loss**
>
>
>
>
> [A4] We would like to highlight that storing prototypical vectors requires significantly less memory than using the additional teacher model as used in distillation approaches. For example, in the ADE20K benchmark, storing DeepLab-V3 (151 classes) requires 58.664 M parameters, while storing 152 prototypical 2048-D vectors (including the unknown cluster) only uses 0.311M parameters. In addition, the computation of $\mathcal{L}_{cluster}$ loss with $\mathbf{p}_c$ is cheaper than a forward pass of the entire network used in distillation. Therefore, in terms of computational cost and memory, our approach remains more efficient compared to knowledge distillation approaches.
>
>
>
>
>
>
>
>
> [Q5] **First line results in Table 1**
>
>
>
>
> [A5] The first line indicates the results of softmax only (mentioned in L307).
>
>
>
>
> [Q6] **The role of conditional structural consistency in fairness**
>
>
>
>
> [A6] As our clarification to Reviewer ReY5 (Q2-A2), the goal of conditional structural consistency is to improve the prediction gap among neighbor pixels, thus, enhancing the smoothness of the predictions. In addition, it helps to increase fairness among classes. It is because this loss helps to clean up the spurious or ambiguous predictions produced by the major classes around minor classes. Therefore, the loss alleviates the dominance of the major groups and improves the accuracy of the minor groups, thus, resulting in fairness that has also been further improved (as illustrated in Table 1 in the paper).
>
>
>
>
> [Q7] **Details of $\mathcal{L}_{class}$ and the final loss combination**
>
>
>
>
> [A7] The fairness loss $\mathcal{L}_{class}$ is calculated based on the predictions produced by the model and the ideal uniform class distribution, as mentioned in L175-176. As our clarification to Reviewer 2Nqo (Q1-A1), the final loss is the summation of the softmax loss with (pseudo) labels, the contrastive clustering loss, the fairness loss, and the conditional structural consistency loss. Our work does not utilize the balance weights among losses. Following common practices [4, 5], we investigate the contributions of our losses and their vital roles in fairness and catastrophic forgetting. We leave the investigation of balanced weights among losses as our future work.
>
>
>
>
> [Q8] **Better measurement of the fairness**
>
>
>
>
> [A8] Please refer to our **Global Response** for our clarification on the performance improvement in major (dominant) and minor (rare) classes.
>
>
>
>
> [Q9] **Typos in Related Work and Equations**
>
>
>
>
> [A9] We have updated typos in related work and equations.
>
>
>
>
> ### References
> [1] Truong et al. Fredom: Fairness domain adaptation approach to semantic scene understanding. CVPR, 2023.
>
>
>
>
> [2] Wang et al. Seesaw loss for long-tailed instance segmentation. CVPR, 2021.
>
>
>
>
> [3] Cui et al. Parametric contrastive learning. ICCV, 2021.
>
>
>
>
> [4] Joseph et al. Towards open world object detection. CVPR, 2021.
>
>
>
>
> [5] Truong et al. Fredom: Fairness domain adaptation approach to semantic scene understanding. CVPR, 2023.

---

> > ### Comment · Reviewer_9fXu · 2023-08-21
> >
> > Thanks for the detailed response. I will keep my rating.

---

> > > ### Author Response · Authors · 2023-08-21
> > > **Thanks For Rating**
> > >
> > > Dear Reviewer 9fXu,
> > >
> > > Thank you very much for your positive feedback and rating.
> > >
> > > Thank you very much,
> > >
> > > Authors

---

### Official Review · Reviewer_2Nqo · 2023-07-05

**Soundness:** 3 good
**Presentation:** 2 fair
**Contribution:** 3 good
**Rating:** 6
**Confidence:** 3

**Summary:**

This paper proposes a Fairness Continual Learning approach to address the continual semantic segmentation problem. Specifically, the fairness continual learning framework is proposed for fairness objective, while a Prototypical Contrastive Clustering loss for reducing catastrophic forgetting. When compared to prior work on continual semantic segmentation, FairCL is shown to improve Final mIoU significantly.

**Strengths:**

1. This paper analyzes several problems that need to be solved in continuous learning in detail, and clearly presents the loss function corresponding to the three objectives from the perspective of formula derivation.
2. Extensive ablation studies are performed which provides insight into the challenges of continual semantic segmentation problem.

**Weaknesses:**

1. The organization of method is not very clear. It's difficult to follow what the final loss function is and how each part of loss is combined.
2. About the Prototypical Contrastive Clustering Loss, I wonder whether pulling all features of unknown classes into one cluster is reasonable. Will learning similar features from categories with significant differences affect subsequent model training?
3. How is the margin $\triangle$ selected?
4. In Sec.3.2, 'a pseudo label where its assigned label is computed based on the closest cluster'. How is the pseudo label computed and utilized?
5. The compare in experiments about network backbone is unfair.  The experiment provides results on two different backbones, but neither backbone was aligned with the previous method such as Deeplab-v3 architecture with ResNet-101 in RCIL.

**Questions:**

I am looking forward to the authors clarifying the concerns listed in the section above:
1. The final loss function and the combination form of each part;
2. More detail explanation of the Prototypical Contrastive Clustering Loss;
3. Comparison in experiment results.

**Limitations:**

Both limitations and potential societal impacts are addressed in the paper. However, it might help to expand the paragraph to discuss some of the concerns listed.

---

> ### Author Rebuttal · Authors · 2023-08-06
>
> ## Response to Reviewer 2Nqo
>
> Dear Reviewer 2Nqo,
>
>
>
>
> We would like to thank the reviewer for your careful read and valuable feedback.
> We are very happy you encourage that our proposed losses are well-formulated and the ablation studies have illustrated the effectiveness of our approach. We appreciate your constructive comments and would like to address these points as follows:
>
>
>
>
>
>
>
>
> [Q1] **Final loss combination**
>
>
>
>
> [A1] In our model, the final training loss is the summation of the softmax loss with labels (and pseudo labels), the contrastive clustering loss, the fairness loss, and the conditional structural consistency loss. In our work, we do not utilize the balance weights among proposed losses. Similar to other common practices [5, 8], we investigate the impact and contribution of our proposed losses and their vital roles in fairness and catastrophic forgetting (as illustrated in the ablation study). We leave the investigation of balanced weights among losses as our future work.
>
>
>
>
> [Q2] **Organization of the paper**
>
>
>
>
> [A2] We would like to clarify our paper organization as follows. Our intention was to follow a clear structured progression to present the background of the fairness continual problem, fairness metric and fairness learning learning (Secs 3.1 and 3.2), followed by our proposed methodology and its properties (Secs 3.3-3.4), results (Sec 4), and discussion of limitations (Sec 5). We aim to present our proposed approach in a logical sequence to ensure a comprehensive understanding of our method. In addition, the combination of training losses has been clarified in the previous answer (Q1-A1). Moreover, it should be highlighted that the reviewers agree our paper is well-written (Reviewer ReY5), well-motivated (Reviewer ReY5), and clear presentation of loss functions corresponding to objectives (Reviewer 2Nqo). We will improve our paper organization promptly based on the comments of the reviewer.
>
>
>
>
>
>
>
>
> [Q3] **Pulling all features of unknown classes into one cluster**
>
>
>
>
> [A3] In our proposed approach, we pull the features of unknown classes into a single cluster because we acknowledge the fact that features of novel classes or unknown classes should aggregate into a region (as proved in [5, 6, 7]) instead of scattering in the feature space where these could be misclassified into the other known classes or be produced as discriminative features (as mentioned in Sec 3.2). Later, in the future step, when the classes of unknown features are known, these will be attracted into their corresponding clusters. We acknowledge that the performing sub-clustering on the unknown features could potentially improve the performance of the model and could be explored as a future research direction of our approach.
>
>
>
>
> [Q4] **The choice of margin $\Delta$**
>
>
>
>
> [A4] In our paper, following the common practice of choosing margin $\Delta$ in prior work [5], we choose $\Delta = 10$. In addition, we also perform an additional ablation study on the ADE20K (100-50) benchmark to investigate the impact of the delta. As shown in the table below, the impact of $\Delta$ does not significantly influence the results due to the minor performance drop.
>
>
>
> Table 1: The effectiveness of choosing $\Delta$ (ADE20K  100-50 Benchmark)
> | $\Delta$ | 0-100 | 101-150 | all | avg |
> | :-------: | :---: | :-----: | :---: | :---: |
> | 5 | 43.04 | 23.84 | 36.68 | 40.23 |
> | **10** | **43.40** | **24.04** | **36.99** | **40.45** |
> | 15 | 42.89 | 23.92 | 36.61 | 40.26 |
>
>
>
>
>
>
>
>
> [Q5] **The use of Pseudo Labels**
>
>
>
>
> [A5] Following the common practice in [1, 2], in addition to the labels of the current task, we utilize the pseudo labels to compute the softmax loss. Moreover, the pseudo labels are used to assign the cluster in Eqns. (8-9).
>
>
>
>
> [Q6] **Fair Comparision**
>
>
>
>
> [A6] The comparisons in our experiments are fair with other baselines. As our clarification to Reviewers jVAT and ReY5, DeepLab-V2 is our typo in the text. We sincerely apologize for this oversight and we are committed to correcting this typo promptly. In our implementation, we use DeepLab-V3 for fair comparisons with PLOP [1], RCIL[2], MiB [3] in our experiments. In addition, we also adopt the SegFormer network as SATS [4].
>
>
>
>
> [Q7] **Discussion of limitations and potential societal impacts**
>
>
>
>
> [A7] Thank you very much for your suggestion. We will extend our discussion about limitations and social impacts in our paper.
>
>
>
>
>
>
>
>
>
>
>
>
> ### References
>
>
>
>
> [1] Douillard et al. Plop: Learning without forgetting for continual semantic segmentation. CVPR, 2021.
>
>
>
>
> [2] Zhang et al. Representation compensation networks for continual semantic segmentation. CVPR, 2022.
>
>
>
>
> [3] Cermelli et al. Modeling the background for incremental learning in semantic segmentation. CVPR, 2020.
>
>
>
>
> [4] Qiu et al. Sats: Self-attention transfer for continual semantic segmentation. Pattern Recognition, 2023.
>
>
>
>
> [5] Joseph et al. Towards open world object detection. CVPR, 2021.
>
>
>
>
> [6] Cen et al. Deep metric learning for open world semantic segmentation. ICCV, 2021.
>
>
>
>
> [7] Li et al. Open-Set Semantic Segmentation for Point Clouds via Adversarial Prototype Framework. CVPR, 2023.
>
>
>
>
> [8] Truong et al. Fredom: Fairness domain adaptation approach to semantic scene understanding. CVPR, 2023.

---

> > ### Comment · Reviewer_2Nqo · 2023-08-19
> >
> > Thanks for your responses,  and I will update my score.

---

> > > ### Author Response · Authors · 2023-08-20
> > > **Thanks For Raising Rating**
> > >
> > > Dear Reviewer 2Nqo,
> > >
> > > Thank you very much for your positive feedback and raising your rating.
> > >
> > > Thank you very much,
> > >
> > > Authors

---

### Official Review · Reviewer_ReY5 · 2023-07-05

**Soundness:** 3 good
**Presentation:** 2 fair
**Contribution:** 3 good
**Rating:** 5
**Confidence:** 4

**Summary:**

The paper explores the continual semantic segmentation problem and highlights the lack of fairness in existing methods. It proposes a novel fairness objective based on class distributions to address the bias on the majority group classes. The paper proposes a novel prototype contrastive clustering loss to address the significant challenges of catastrophic forgetting as well as for handling unknown classes to solve the background shift problem. Furthermore, a conditional structural consistency loss is proposed to constrain the structure of the predicted segmentation. The method has achieved improvements over the existing benchmarks on ADE20K, Cityscapes and Pascal VOC datasets.

**Strengths:**

1. The paper is well written and the motivation is good. The problem formulation is nicely done.

2. The paper points out the issue of maintaining fairness in continual segmentation systems. It is a very significant contribution to the field since all existing methods primarily focussed on reducing forgetting and addressing the background shift.  The fairness problem is well-defined in the paper along with the fairness metric which is used to obtain the error-rate among the classes.

3. The novel prototype contrastive clustering loss represents the semantic segmentation pixels for different classes in the latent space and groups the features of unknown classes into the same cluster. In addition to handling unlabelled classes, it is interesting to see that the clustering loss also serves as an upper bound of the knowledge distillation approach and thus reduces the forgetting of old classes.

4. The proposed method FairCL seems to be quite effective on several settings across multiple datasets with different backbones.

**Weaknesses:**

1. Although the fairness loss is improving the mIoU values for different settings on ADE20K, it is not clear if the improvement in mIoU is coming from the minority classes. While the main motivation of the paper is to ensure fairness among the classes, the ablation experiment does not completely justify this. The individual contribution of the fairness loss (L_class) should be stated since Fairness in Continual Learning is the main claim of the authors. Further analysis of the improvements in mIoU values separately for majority and minority classes on at least two datasets will be more convincing. Also, the fairness loss may be evaluated in a non-incremental setting (let's say for the first step only), and then the minority class mIoU values should be better; otherwise, the performance is also affected by forgetting, and the contribution of the fairness loss is not clear.

2. The motivation for learning conditional structural consistency loss is not clear and should be discussed to give more intuition. Also, some qualitative analysis of prediction consistency will be good to show the effectiveness of the structural consistency loss.

3. Results from Pascal VOC are not convincing. For the 10-1 setting, the forgetting is quite high compared to methods like RCIL. Overall, the improvement looks very insignificant. Experiments on more settings of Pascal VOC similar to previous papers like PLOP and RCIL will be good. Also, the comparison of the SegFormer backbone with the other existing methods using DeepLabV2 is not fair. If there is any specific reason for the method not being so effective for Pascal VOC (as it looks effective for ADE20K and Cityscapes), it should be analyzed.

**Questions:**

1. Inconsistent results: The results for the ADE20K 100-50 setting are different in table 1 and table 3. In table 1 for ablation, the overall mIoU is 34.64 for DeepLab-V2 backbone, while in table 3, the overall mIoU is 36.99. The mIoU values are the same for the SegFormer backbone though. This should be clarified since the ablation experiment is claiming the contributions of the different losses.

2. While it is good to see the results with both DeepLab-V2 and SegFormer backbones, the backbone networks of the compared methods should be mentioned for more clarity and fair comparison.

3. The comparison does not include a well performing baseline method - SSUL (from NeurIPS 21). If there is any specific reason for not comparing with SSUL, it should be discussed in the paper.

4. See weaknesses 1, 2, 3.

**Limitations:**

/

---

> ### Author Rebuttal · Authors · 2023-08-06
>
> ## Response to Reviewer ReY5
>
>
>
>
> Dear Reviewer ReY5,
>
>
>
>
> We would like to thank the reviewer for your careful read and valuable feedback.
> We appreciate that you think our paper is well-motivated and well-written, the proposed approach is well-formulated and novel, and our theoretical analysis of the upper bound of our proposed approach is significant. We appreciate your constructive comments and would like to address these points as follows:
>
>
>
>
> [Q1] **The performance improvement in minor and major classes**
>
>
>
>
> [A1] Please refer to our **Global Response** for our clarification on the performance improvement in minor and major classes in both continual learning and non-incremental settings.
>
>
>
>
>
>
>
>
> [Q2] **Motivation for learning conditional structural consistency**
>
>
>
>
> [A2] The goal of conditional structural consistency is to improve the prediction gap among neighbor pixels, thus, enhancing the smoothness of the predictions. In addition, it helps to increase fairness among classes. It is because this loss helps to clean up the spurious or ambiguous predictions produced by the major classes around minor classes. Therefore, the loss alleviates the dominance of the major groups and improves the accuracy of the minor groups, thus, resulting in fairness that has also been further improved (as illustrated in Table 1 in the paper).
>
>
>
>
>
>
>
>
> [Q3] **Results from Pascal VOC**
>
>
>
>
> [A3] In our VOC experiments, similar to [1, 2], we choose the continual learning settings of 10-1 and 15-1 since these two settings illustrate more the impact of forgetting and the effectiveness of the method to the forgetting problem. In addition, we have observed the performance improvement of our approach on Pascal VOC is less significant than the ADE20K and Cityscapes because of the minor bias in Pascal VOC. In particular, the data distributions of these datasets are visualized in Figure 1 in the rebuttal file (the PDF file of the rebuttal is attached in Global Response). We calculated the entropy value of the data distributions that illustrate the balance level of the datasets (the higher value of entropy, the more balance the dataset as the data distribution tends to be more uniform). Then, based on the data distributions and entropy values, we observe the data distributions of ADE20K and Cityscapes suffer more bias than the Pascal VOC since the entropy value of Pascal VOC (H = 0.81) is higher than ADE20K (H = 0.69) and Cityscapes (H = 0.62). Thus, ADE20K and Cityscapes suffer severe fairness issues compared to Pascal VOC. Our approach aims to improve the fairness of the model. Therefore, on the more severe bias datasets (ADE20K and Cityscapes), our approach performs more significantly.
>
>
>
>
> [Q4] **Inconsistent Results**
>
>
>
>
> [A4] Upon further investigation, we have discovered that the inconsistencies between Table 1 and Table 3 in the paper were a result of typographical errors that occurred during the typing process. We have updated the correct results in Table 1 (in the paper), consistent with our results in Table 3 (in the paper). We would like to offer our sincere apologies for this oversight.
>
>
>
>
>
>
>
>
> [Q5] **Fair Comparision**
>
>
>
>
> [A5] The comparisons in our experiments are fair with other baselines. As clarified to Reviewer jVAT, DeepLab-V2 is our typo in the text. We sincerely apologize for this oversight and are committed to promptly correcting this typo. In our implementation, we use DeepLab-V3 (with ResNet 101) for fair comparisons with PLOP [1], RCIL [2], and MiB [3] in our experiments. Similarly, we adopt SegFormer as SATS [4] for fair comparisons.
>
>
>
> [Q6] **Result of SSUL**
>
>
>
>
> [A6] We have compared our results with SSUL [5] in the table below. Our paper achieve a competitive performance compared to SSUL on the ADE20K benchmarks.
>
>
>
> Table 1: Comparisons with SSUL on ADE20 Benchmarks
> | Method | | 100-50 | | | 50-50 | | | 100-10 | |
> | :- | :-: | :-: | :-: | :-: | :-: | :-: | :-: | :-: | :-: |
> | | 0-100 | 101-150 | all | 0-50 | 51-150 | all | 0-100 | 101-150 | all |
> | SSUL | 41.28 | 18.02 | 33.58 | 48.38 | 20.15 | 29.56 | 40.20 | 18.75 | 33.10 |
> | FairCL + DeepLab-V3 | 43.40 | 24.04 | 36.99 | 49.65 | 26.84 | 34.55 | 41.73 | 20.36 | 34.65 |
> | FairCL + SegFormer | 43.56 | 25.46 | 37.56 | 49.62 | 27.78 | 35.15 | 42.21 | 21.91 | 35.49 |
>
>
>
>
>
>
>
>
> ### References
>
>
>
>
> [1] Douillard et al. Plop: Learning without forgetting for continual semantic segmentation. CVPR, 2021.
>
>
>
>
> [2] Zhang et al. Representation compensation networks for continual semantic segmentation. CVPR, 2022.
>
>
>
>
> [3] Cermelli et al. Modeling the background for incremental learning in semantic segmentation. CVPR, 2020.
>
>
>
>
> [4] Qiu et al. Sats: Self-attention transfer for continual semantic segmentation. Pattern Recognition, 2023.
>
>
>
>
> [5] Cha et al. SSUL: Semantic segmentation with unknown label for exemplar-based class-incremental learning. NeurIPS, 2021.

---

> > ### Comment · Reviewer_ReY5 · 2023-08-18
> > **Response for the authors**
> >
> > Thank you for your detailed responses, which addresses most of my concerns. I will maintain my score.

---

> > > ### Author Response · Authors · 2023-08-18
> > > **Thanks For Rating**
> > >
> > > Dear Reviewer ReY5,
> > >
> > > Thank you very much for your positive feedback and rating. We are committed to updating our paper based on the reviewers' suggestions.
> > >
> > > Thank you very much,
> > >
> > > Authors

---

### Official Review · Reviewer_jVAT · 2023-07-24

**Soundness:** 3 good
**Presentation:** 4 excellent
**Contribution:** 3 good
**Rating:** 7
**Confidence:** 4

**Summary:**

The paper focuses on the semantic segmentation problem in the context of continual learning. In the past, the emphasis was mainly on addressing catastrophic forgetting and background shift issues. However, this paper goes beyond that and investigates the balance problem between different object categories. It introduces novel approaches for prototype contrastive learning and structure-consistency loss functions, which result in a new SOTA performance. The contributions of the paper can be summarized into three main parts:

i. It introduces a novel evaluation metric to measure the model's balance across different classes.

ii. It proposes a new contrastive learning loss to address the issue of forgetting in continual learning.

iii. The paper conducts extensive experiments to demonstrate the effectiveness and rationality of the proposed methods.

**Strengths:**

+ The paper considers multiple perspectives to improve fairness among classes of the model.
+ The paper designs various loss functions to enhance the segmentation performance of the model under continual learning.
+ The supplementary materials of the paper include an excellent presentation that makes it easier for readers to understand the superiority of the model.

**Weaknesses:**

+ The paper lacks experiments specifically focused on the upper bound design for continual learning.
+ The proposed inter-class fairness in the paper seems to be addressing the semantic segmentation problem rather than the main focus of continuous learning. However, this issue has already been well discussed in [1].
[1] Truong, T. D., Le, N., Raj, B., Cothren, J., & Luu, K. (2023). Fredom: Fairness domain adaptation approach to semantic scene understanding. In Proceedings of the IEEE/CVF Conference on Computer Vision and Pattern Recognition (pp. 19988-19997).

**Questions:**

As far as I know, the current research on semantic segmentation is primarily conducted using DeepLab v3 as the framework. Why did the paper choose to experiment with DeepLab v2 instead?

**Limitations:**

The paper discusses its own limitations in the experimental section. The paper does not have any negative social impact.

---

> ### Author Rebuttal · Authors · 2023-08-06
>
> ## Response to Reviewer jVAT
>
> Dear Reviewer jVAT,
>
> We thank the reviewer for your careful read and valuable feedback.
> We are pleased you think of our proposed problem and approach as our strengths and significant contributions. We appreciate your constructive comments and would like to address these points as follows.
>
> [Q1] **The upper bound design for continual learning**
>
> [A1] We have illustrated the upper-bound results of our method, where the upper-bound results are the performance of the model trained on the joint data tasks. As shown in the below tables, the gap in performance of our method with the upper bound remains lower than other approaches in all three benchmarks.
>
>
>
>
> Table 1: ADE20K Dataset
>
>
>
>
> | Method | | 100-50 | | | 50-50 | | | 100-10 | |
> | :------------------ | :---: | :-----: | :---: | :---: | :----: | :---: | :---: | :-----: | :---: |
> | | 0-100 | 101-150 | all | 0-50 | 51-150 | all | 0-100 | 101-150 | all |
> | Upper Bound | 44.30 | 28.20 | 38.90 | 51.10 | 32.80 | 38.90 | 44.30 | 28.20 | 38.90 |
> | MiB + AWT | 40.90 | 24.70 | 35.60 | 46.60 | 26.85 | 33.50 | 39.10 | 21.28 | 33.20 |
> | SATS | - | - | - | - | - | - | 41.42 | 19.09 | 34.18 |
> | FairCL + DeepLab-V3 | 43.40 | 24.04 | 36.99 | 49.65 | 26.84 | 34.55 | 41.73 | 20.36 | 34.65 |
> | FairCL + SegFormer | 43.56 | 25.46 | 37.56 | 49.62 | 27.78 | 35.15 | 42.21 | 21.91 | 35.49 |
>
>
>
>
>
>
>
>
> Table 2: Pascal VOC Dataset
>
>
>
>
> | Method | | 15-1 | | | 10-1 | |
> | :------------------ | :--: | :---: | :--: | :--: | :---: | :--: |
> | | 0-15 | 16-20 | all | 0-10 | 11-20 | all |
> | Upper Bound | 79.8 | 72.4 | 77.4 | 78.4 | 76.4 | 77.4 |
> | RCIL | 70.6 | 23.7 | 59.4 | 55.4 | 15.1 | 34.3 |
> | FairCL + DeepLab-V3 | 72.0 | 22.7 | 60.3 | 42.3 | 25.6 | 34.4 |
> | FairCL + SegFormer | 73.5 | 22.8 | 61.5 | 57.1 | 14.2 | 36.6 |
>
>
>
>
>
>
>
>
> Table 3: Cityscapes Dataset
>
>
>
>
> | Method | 11-5 | 11-1 | 1-1 |
> | :------------------ | :---: | :---: | :---: |
> | | all | all | all |
> | Upper Bound | 79.30 | 79.30 | 79.30 |
> | RCIL | 64.30 | 63.00 | 48.90 |
> | FairCL + DeepLab-V3 | 66.96 | 66.61 | 49.22 |
> | FairCL + SegFormer | 67.85 | 67.09 | 55.68 |
>
>
>
>
>
>
>
>
> [Q2] **Comparison to [1]**
>
>
>
>
> [A2] In our related work section, we discussed the FREDOM paper [1] in L97-99. As the reviewer highlighted, the focus of [1] and our paper remains different since our work concentrates on the fairness problem in the context of continual learning. Meanwhile, FREDOM addresses fairness in the context of learning from the entire data at once under the domain adaptation setting.
>
>
>
>
> [Q3] **Clarification of Segmentation Network**
>
>
>
>
> [A3] DeepLab-V2 is our typo in the text. In our implementation, we use ***DeepLab-V3*** for fair comparison in our experiments (similar to PLOP [2] RCIL [3], MiB [4]). We sincerely apologize for this oversight and assure you that we are committed to correcting this typo promptly.
>
>
>
>
> ### References
>
>
>
>
> [1] Truong et al. Fredom: Fairness domain adaptation approach to semantic scene understanding. CVPR, 2023.
>
>
>
>
> [2] Douillard et al. Plop: Learning without forgetting for continual semantic segmentation. CVPR, 2021.
>
>
>
>
> [3] Zhang et al. Representation compensation networks for continual semantic segmentation. CVPR, 2022.
>
>
>
>
> [4] Cermelli et al. Modeling the background for incremental learning in semantic segmentation. CVPR, 2020.

---

> > ### Comment · Reviewer_jVAT · 2023-08-18
> >
> > Thank the authors for their responses. I don't have any more questions. I hope the mentioned issues can be addressed in the future version, and I will raise my score to 7.

---

> > > ### Author Response · Authors · 2023-08-18
> > > **Thanks For Raising Rating**
> > >
> > > Dear Reviewer jVAT,
> > >
> > > Thank you very much for your positive feedback and raising your rating. We are committed to updating our paper based on the reviewers' suggestions.
> > >
> > > Thank you very much,
> > >
> > > Authors

---

### Author Rebuttal · Authors · 2023-08-06

## Global Response


We thank all the reviewers for their careful read and valuable feedback. Reviewer 9fXu and Reviewer jVAT ***weakly accept*** our paper; Reviewer ReY5 considers our work a ***borderline accept***; and Reviewer 2Nqo considers ***a borderline reject***. Our implementation will be published for research reproducibility. We are very happy the reviewers encourage that ***our work is well-motivated*** (Reviewer ReY5), ***our paper is well-written*** (Reviewers ReY5, 2Nqo), ***our method is novel*** (Reviewers jVAT, ReY5, 9fXu), and ***our approach achieves strong performance*** (Reviewers ReY5, 2Nqo, 9fXu). We have updated the typos in our paper.








On the constructive side, Reviewer ReY5 and Reviewer 9fXu noted that better fairness measurement will be beneficial to illustrate the effectiveness and the contribution of the proposed method to fairness improvement. Therefore, as suggested by the reviewers, to illustrate the performance improvement of our proposed method in major and minor classes, we include the results of the mIoU (all) and the STD among IoUs of the major group and the minor group on the ADE20K 100-50 (Table 1) and Cityscapes 11-5 (Table 2) benchmarks. As shown in the table below, our proposed approach has improved the performance of both major and minor groups. Thus, these results illustrated that the performance improvement in mIoU also comes from the minority classes. It helps to enhance the mIoU performance and reduce the STD in both major and minor classes, thus, improving the fairness of the model predictions.








Table 1: ADE20K 100-50 Benchmark
| Backbone | $\mathcal{L}_{cluster}$ | $\mathcal{L}_{class}$ | $\mathcal{L}_{cons}$ | Major | Major | Minor | Minor |
| - | :-: | :-: | :-: | :-: | :-: | :-: | :-: |
||||| mIoU | STD | mIoU | STD |
| DeepLab-V3 | yes | | | 48.78 | 18.12 | 25.13 | 21.12 |
| DeepLab-V3 | yes | yes | | 48.89 | 17.87 | 27.24 | 20.76 |
| DeepLab-V3 | yes | yes | yes | 50.11 | 17.46 | 30.52 | 20.43 |




Table 2: Cityscapes 11-5 Benchmark
| Backbone | $\mathcal{L}_{cluster}$ | $\mathcal{L}_{class}$ | $\mathcal{L}_{cons}$ | Major | Major | Minor | Minor |
| - | :-: | :-: | :-: | :-: | :-: | :-: | :-: |
| | | | | mIoU | STD | mIoU | STD |
| DeepLab-V3 | yes | | | 87.44 | 9.25 | 53.37 | 16.72 |
| DeepLab-V3 | yes | yes | | 88.29 | 8.85 | 55.72 | 13.39 |
| DeepLab-V3 | yes | yes | yes | 89.20 | 8.41 | 56.70 | 11.96 |
















Similarly, to illustrate the effectiveness and robustness of our method in the non-incremental setting. We report our results after the first learning step on the ADE20K 100-50 (Table 3) and Cityscapes 11-5 (Table 4) benchmarks. Our proposed fairness approach has also contributed to the performance improvement of both major and minor classes in non-incremental settings. The comparison table of major and minor groups in the first step is illustrated below.








Table 3: ADE20K 100-50 Benchmark (Non-incremental Setting)
| Backbone | $\mathcal{L}_{cluster}$ | $\mathcal{L}_{class}$ | $\mathcal{L}_{cons}$ | Major | Major | Minor | Minor |
| - | :-: | :-: | :-: | :-: | :-: | :-: | :-: |
| | | | | mIoU | STD | mIoU | STD |
| DeepLab-V3 | yes | | | 49.07 | 18.35 | 35.45 | 19.08 |
| DeepLab-V3 | yes | yes | | 49.17 | 18.31 | 35.57 | 18.24 |
| DeepLab-V3 | yes | yes | yes | 50.04 | 17.92 | 37.88 | 18.19 |




Table 4: Cityscapes 11-5 Benchmark (Non-incremental Setting)
| Backbone | $\mathcal{L}_{cluster}$ | $\mathcal{L}_{class}$ | $\mathcal{L}_{cons}$ | Major | Major | Minor | Minor |
| - | :-: | :-: | :-: | :-: | :-: | :-: | :-: |
| | | | | mIoU | STD | mIoU | STD |
| DeepLab-V3 | yes | | | 87.64 | 8.96 | 53.78 | 17.16 |
| DeepLab-V3 | yes | yes | | 88.56 | 8.70 | 56.45 | 13.46 |
| DeepLab-V3 | yes | yes | yes | 89.52 | 8.08 | 57.87 | 12.08 |

---

### Decision · Program_Chairs · 2023-09-21

**Decision:**

Accept (poster)

**Comment:**

The paper describes an approach for fairness continual learning applied to semantic segmentation. The paper provided both theoretical insights on the proposed method and convincing experimental results proving the effectiveness of the proposed approach (which outperforms competitors). The reviewers mostly asked clarifications about the method (fairness objective, hyperparameters) and a few experimental results (some missing ablation). The authors provided all the necessary clarifications and few additional results. All reviewers agree that the paper can be accepted.